# Characteristics of Occupational Exposure to Diesel Engine Exhaust for Shipyard Transporter Signal Workers

**DOI:** 10.3390/ijerph17124398

**Published:** 2020-06-18

**Authors:** Jungah Shin, Boowook Kim, Hyoung-Ryoul Kim

**Affiliations:** 1Department of Research for Occupational Health, Institute of Occupation and Environment, Incheon 21417, Korea; sja2014@kcomwel.or.kr (J.S.); kimbw@kcomwel.or.kr (B.K.); 2Department of Occupational and Environmental Medicine, College of Medicine, The Catholic University of Korea, Seoul 06591, Korea

**Keywords:** diesel engine exhaust, elemental carbon, occupational exposure, transporter signal work

## Abstract

*Background*: Workers performing signal work for a heavy-duty shipyard transporter are exposed to diesel engine exhaust (DEE), which is classified as a Group 1 carcinogen by the International Agency for Research on Cancer. Here, we evaluate DEE exposure levels for workers engaged in shipyard transporter signal work through measurement of respirable elemental carbon (EC), organic carbon (OC), and total carbon (TC), and identify the factors affecting exposure. *Methods*: Sixty signal workers were selected, and measured samples were analyzed by thermo-optical transmittance. *Results*: The mean EC exposure level of a transporter signal worker was 4.16 µg/m^3^, with a range of 0.69 to 47.81 µg/m^3^. EC, OC, and TC exposure levels did not show statistically significant differences for individual variables except the measurement date. This was influenced by meteorological factors such as wind speed, and it was confirmed that the work position, number carried, and load capacity in the multiple regression analysis after minimizing the meteorological effects were factors influencing the EC exposure level of the signalman. *Conclusions*: Meteorological conditions influenced DEE exposure of transporter signal workers who work outdoors. The mean EC exposure level was not high, but exposures to high concentrations of EC were recorded by meteorological factors.

## 1. Introduction

A shipyard transporter is a large vehicle used to move parts of a ship, such as steel sections or blocks and equipment. Shipyard transporters are also used in power plants or factories to move heavy industrial equipment or automobiles. Workers use signals to avoid colliding with other equipment or vehicles. The worker in charge of the transporter uses a whistle to alert the driver to possible collisions during loading and unloading. Simultaneously, a signal worker moves alongside on a bicycle. Three signal workers are involved in moving each transporter. Unlike the driver in the cabin of the transporter, they work directly near the vehicle’s exhaust pipe and are therefore more susceptible than the driver to the effects of diesel engine exhaust (DEE).

DEE was classified as a Group 1 carcinogen that causes lung cancer by the International Agency for Research on Cancer in 2012 [1]. Incomplete combustion of diesel fuel generates complex chemicals in the gas and liquid phase along with diesel particulate matter (PM) [1,2,3]. DEE can be divided into gas, liquid, and particulate fractions. The gas and liquid fractions are mainly NO, NO_2_, CO, CO_2_, SO_2_, and other low-molecular-weight volatile organic compounds such as benzene, 1,3-butadiene, and aldehydes. The PM fraction is composed primarily of elemental carbon (EC), organic carbon (OC), ash, polycyclic aromatic hydrocarbons, and metals, and most are smaller than 1 µm in diameter [1,2,3,4,5]. The composition of DEE can vary depending on the type and age of the engine, driving cycle, working conditions, fuel, emission control system, and type of engine oil. In the case of a diesel engine that does not have an emission control device, the composition of the DEE is known to contain more particulate matter (PM). High concentrations of PM are associated with incomplete combustion of diesel fuel, particularly under high-load conditions. Under low-load conditions, PM concentrations are relatively low and increased adsorption of organic matter with particles has been observed [6,7,8].

Outdoor EC and OC concentrations can be affected significantly by environmental and meteorological factors. Temperature, humidity, and wind affect background concentrations and exposure [9,10]. PM tends to accumulate in stagnant air at weak wind speeds and in a stable atmosphere [11]. Strong wind dilutes and mixes the atmosphere, resulting in lower PM concentrations, such as PM2.5 (PM ≤ 2.5 µm in diameter) [12,13].

Several studies have evaluated occupational exposure to DEE using various surrogates including EC. EC exposure levels have been reported for drivers, mechanics, miners, train crews, construction workers, dockworkers, and household-waste collectors [2,14,15,16]. However, there is a lack of data for shipyard transporter signal workers. Regulations on diesel engine emissions now stringently control DEE. Laws apply to on-road light- and heavy-duty vehicles and off-road vehicles; however, certain heavy-duty engines, trains, or ships are often exempt [1]. Diesel engine vehicles operating under high-load conditions of over 100 t, as in the case of a shipyard transporter, are often not subjected to these regulations.

We evaluated the individual DEE exposure levels of workers while performing transporter signal duties in a shipyard, using respirable EC, OC, and total carbon (TC, EC + OC), as representative surrogates. We identified work characteristics and factors influencing exposure to EC, OC, and TC to examine DEE exposure characteristics in shipyard transporter signal work.

## 2. Subjects and Methods

### 2.1. Brief Description of Shipyard Transporter Signal Worker

A signal worker transmits signals to the transporter during the block-loading process while en route to the designated destination and when unloading the block. The tasks of the signal worker can be categorized into signalman, forklift driver, and uniloader driver. A signalman informs the driver, who has a limited range of vision, about surrounding obstacles and situations during the movement by radio or whistle, hand signals, or signal rods. When a transporter moves, two workers will cover the front left and right of the transporter, while one observes the rear. The destination may be a painting booth, blasting booth, or an external assembly site. Upon unloading, the signal worker informs the driver of the transporter height and location to support the safe positioning of the ship block on trestles or stools.

The forklift and uniloader drivers place trestles or stools for support, position the arriving block at the designated place, and then replace the frame or stool after the block is moved.

The type of transporter varies by the load, which can range from 100 to 1000 t. A total of nine vehicles participated in our study, from 350 to 1000 t, including four transporters with loading capacities (the maximum load that can be moved) of less than 500 t and five greater than 500 t. We included older and newer transporters for each loading capacity. Other variables included transporter body weight (t), body length (m), and how many items were moved on the day of measurement (Table 1). The speed of a transporter is approximately 10 km/h empty and 5–6 km/h with a maximum load. Driver’s seats are located at both the front and rear of a transporter. When the transporter needs to go in the opposite direction to the existing direction, the transporter itself does not reverse. Instead, the driver moves from one seat to the other as the signaling workers trade places. One of the two front signalmen moves to the rear, giving a signal at the front, and the existing front signalman gives a signal from the back. The position depends on the day of the work, but the front and rear parts change frequently depending on the direction of movement of the transporter; the ratio of front work and rear work therefore cannot be divided equally. The front signal worker spends approximately 60% of a day at the front and 40% at the rear. As the transporter signal worker uses a radio and whistle to signal, they do not use respirators when working, and there are two or three members per team. The exhaust port is in the middle of the transporter side, and there is no separate emission-reduction device. Most transporter signal work takes place outdoors.

### 2.2. Exposure Assessment Strategy

We collected data over three days at a shipyard in Gyeongsangnam-do, South Korea, in May and June of 2019. To minimize the influence of the sources that are not related to transporter signal work, measurements were performed at night when no other equipment or vehicles were in operation. Two teams transporting blocks using a transporter were selected, and respiratory EC and OC levels were measured as indicators of DEE. The total number of workers participating in the study was 60, with 48 transporter signalmen, 5 uniloader drivers, and 7 forklift drivers. The working hours began at 8 p.m. and finished at 8 a.m. the next morning. Measurements were conducted from 7:30 to 8:30 p.m. until 5 to 7 a.m. the following day. During the measurement study periods, 1 h for lunch (12 to 1 a.m.) and 1 h for a break (3:30 to 4:30 a.m.) were excluded from the measurements.

Samples from the workers’ breathing zone were measured. We used 37 mm quartz filters preloaded in three-piece clear plastic cassettes (cat. no. 225–401, SKC Inc., Eighty Four, PA, USA) with a respirable dust cyclone (GK2.69, BGI Inc., Waltham, MA, USA) using a personal sampling pump (AirChek XR5000, SKC Inc., Eighty Four, PA, USA). A GK2.69 cyclone has a 50% cut point of 4 μm at a flow rate of 4.2 L/min. Therefore, according to the manufacturer’s recommendation, a pump flow rate of 4.2 L/min was used. A flow meter (Bios defender 510, Bios International Co., Butler, NJ, USA) aided the calibration of the pumps before and after use.

The measured samples were analyzed using a Lab OC-EC aerosol analyzer (Sunset Laboratory Inc., Tigard, OR, USA) according to measurement analysis method 5040 of the National Institute for Occupational Safety and Health [17]. Analysis took place at the Institute of Occupation and Environment, Korea Workers’ Compensation and Welfare Service, which participates in the American Industrial Hygiene Association Proficiency Analytical Testing program. Performance evaluation samples (PES-1 {low concentration}; PES-5 {mid}; PES-10 {high}, Sunset Laboratory Inc., Tigard, OR, USA) were purchased and analyzed with the samples to confirm the accuracy of sample analysis. The limit of detection of the analyzer was 0.15 μg/cm^2^, and none of the samples measured in this study were undetected.

### 2.3. Meteorological Data

Meteorological factors, including temperature (℃), humidity (%), wind speed (km/h), and prevailing winds, all of which can affect transporter signal work, were supplied by the Automatic Weather System of the Korea Meteorological Administration. We documented temperature, humidity, wind speed, and the prevailing wind each hour from 6 p.m. to 6 a.m. on each measurement day. The arithmetic means of temperature, humidity, and wind speed were used for statistical analysis to confirm their effects.

### 2.4. Data Analysis

We investigated several factors affecting EC, OC, and TC exposure levels in the air the workers breathed during their signal work. Measurement month (May vs. June), department (part 1 and 2), tasks of signal work (signalman, uniloader driver, forklift driver), signal work position (front vs. rear), transporter load capacity (<500 t vs. <1000 t vs. ≥1000 t), transporter age (<10 years vs. ≥10 years), number carried (<25 vs. ≥25), and meteorological conditions (wind speed, temperature, humidity) were recorded as they are expected to have distinct exposure characteristics.

The normality of time-weighted average data was verified by Kolmogorov–Smirnov analysis at a 5% significance level. However, we confirmed that EC, OC, and TC did not show normal distributions (*p* < 0.05). For statistical analysis, we converted all data to a natural logarithm value. The results of Kolmogorov–Smirnov analysis showed that exposure levels of EC (*p* = 0.081), OC (*p* = 0.200), and TC (*p* = 0.092) showed a lognormal distribution. Geometric mean (GM) and geometric standard deviation (GSD), and minimum and maximum values were calculated using descriptive statistics. With the logarithmic exposure-level values, one-way analysis of variance (ANOVA) was used to examine whether there was a significant difference in the mean concentration of EC, OC, and TC according to the occupational and environmental variables. Multiple regression analysis was performed to identify the main factor affecting EC exposure levels and Pearson’s correlation analysis was performed to determine the correlation between logarithmic EC, OC, and TC exposure levels. All statistical analysis was performed using SPSS 17.0 (IBM SPSS statistics, Chicago, IL, USA).

## 3. Results

### 3.1. EC, OC, and TC Exposure Levels

The exposure levels to respirable EC, OC, and TC were assessed for 60 workers who performed transporter signal work. Table 2 presents the exposure levels to EC, OC, and TC in the transporter signal work for each variable, and in the 60 samples submitted, both EC and OC were above the detection limit. The EC exposure level in the entire signal work was 0.69–47.91 μg/m^3^, indicating a wide range of exposure, and GM (GSD) was 4.16 (2.76) μg/m^3^. The exposure levels for OC and TC were 4.66–49.90 and 7.28–97.71 μg/m^3^, respectively, and GM (GSD) was 15.81 (1.73) and 21.09 (1.85) μg/m^3^. The exposure levels of EC, OC, and TC according to variables such as the department, task, signalman position, number carried, transporter age, and load capacity of carriers did not show statistically significant differences (*p* > 0.05). We confirmed that the measurement month affected exposure levels (*p* < 0.05). The GM and GSD values in May and June were 13.39 (2.90) and 2.91 (1.99) μg/m^3^, respectively, for the EC exposure level for each measurement month. These data indicate that the exposure level of signal work measured in May was approximately 4.6 times higher than that measured in June. Similarly, for OC exposure, the GM and GSD values in May and June were 26.07 (1.59) and 13.57 (1.62) μg/m^3^, respectively, and for TC, the GM and GSD were 41.23 (1.91) and 17.20 (1.54) μg/m^3^, respectively, all indicating that the exposure level measured in May was higher than that measured in June. The exposure levels for EC, OC, and TC were similar for each department performing signal work. The EC exposure levels identified for each task were in the order of signalman, uniloader driver, and forklift driver, and GM (GSD) values were 4.49 (2.52), 4.03 (4.04), and 2.50 (3.85) μg/m^3^, respectively, indicating GSD was considerably large even for the same task. For the cases of the uniloader driver and forklift driver, GSD was three times or higher and, theoretically each task could not be called a similar exposure group. The highest EC exposure for each task was 47.81, 45.73, and 42.04 μg/m^3^ in the signalman, uniloader driver, and forklift driver, respectively, and all were measured in May. The GM (GSD) of signalman EC according to the working position was 3.84 (2.51) and 5.97 (2.42) μg/m^3^ in the front and rear, respectively, indicating that the GM concentration of the rear worker was slightly higher. However, the difference was not statistically significant (*p* = 0.115).

The difference in the mean exposure levels of EC, OC, and TC among the cases when the number carried was <25, and when it was ≥25 on the day of measurement it was not statistically significant (*p* = 0.454). The mean exposure level according to transporter age did not show a statistically significant variation in the case of OC and TC. However, the mean exposure level of new transporters tended to be high. In the case of EC, there was a statistically significant difference between the mean exposure level of the signalman working with a transporter <10 years old and exposure level with a transporter ≥10 years old. The mean EC exposure level with a transporter <10 years old was more than twice as high as that with a transporter ≥10 years old (*p* = 0.011).

### 3.2. Meteorological Influences

The differences in EC, OC, and TC exposure levels by date of measurement of the transporter signal work are shown in Figure 1. The EC, OC, and TC exposure levels appear to be affected by temperature (°C), humidity (%), wind speed (km/h), and prevailing winds by measurement date. On the first day of measurement, the temperature, humidity, and wind speed were 16.9 °C, 54.3%, and 4.6 km/h, respectively. On the second day, the values were 17.8 °C, 86.7%, and 19.1 km/h, and on the third day, the values were 17.9 °C, 86.4%, and 11.8 km/h. The measured temperature values of all three days were similar, but humidity on the first day was only about 63% of the humidity of the second and third days. The wind speed on the first day was 4.6 km/h, which is slower than the transporter’s driving speed, and four times and 2.5 times slower than the wind speed on the second and third days, respectively. In contrast, the prevailing wind on the first day was WNW, NW, and NNE, but ENE, NE, and E prevailed on the second, and ENE, ESE, and NW prevailed on the third. The GMs (GSD) of the signalman’s EC exposure level were 10.95 (2.76), 3.12 (1.74), and 3.57 (2.21) μg/m^3^, respectively, showing the highest value on the first day, a day with low humidity and slow wind speed. The daily EC exposure level of the signalman showed a statistically significant difference (*p* = 0.003). Furthermore, the signalman’s OC and TC exposure levels showed the highest value on the first day, and the daily EC exposure levels was statistically significantly different (*p* < 0.05). Although not statistically confirmed, the exposure levels of EC, OC, and TC were also the highest on the first day when the humidity was low and the wind speed was slow, in the case of forklift drivers and uniloader drivers. A wind speed of 4.6 km/h on the first day corresponded to light air (class 1) on the Beaufort wind scale, and speeds of 19.1 and 11.8 km/h on the second and third days corresponded to a gentle breeze (class 3) and light breeze (class 2), respectively. In 2019, the number of days corresponding to light air with a mean daily wind speed per hour of ≤1.5 m/s in the study area was 84, accounting for 23.0% of the year. We classified a total of 206 days (56.4%) as a light breeze with ≤3.3 m/s and a total of 38 days (10.4%) as a gentle breeze (Appendix A).

### 3.3. Multiple Linear Regression Analysis

The exposure level of EC, the representative surrogate of DEE, is considered to be significantly affected by weather conditions such as wind speed. In order to identify variables that influence the exposure level of natural log-transformed EC excluding the effect of wind, multiple regression analysis was performed using the results measured in June, when the effect of wind was relatively small. The results are shown in Table 3.

Six variables were included in the multiple regression analysis, i.e., measurement date, department, signal work position, transporter age, number carried, and load capacity. The variables included in the final model for estimating the exposure level were selected by the backward elimination method. The selected variables are signal work position, number carried, and load capacity (adjusted R^2^ = 0.480, *p* < 0.001, Durbin−Watson = 2.018).

### 3.4. Relationship between EC, OC, and TC

The mean ratio of OC exposure and EC exposure (OC/EC) was 5.27 (range: 0.88–25.14). The OC/EC ratios differed according to EC exposure level. The mean was 1.34 when the EC concentration was greater than 10 μg/m^3^ (range: 0.88–2.36), and 6.16 when the EC concentration was 10 μg/m^3^ or lower (range: 1.10–25.14), indicating that the higher the EC exposure level, the lower the ratio of OC/EC. The OC/EC ratio by EC concentration tended to be similar to the overall mean in the group-specific results, as confirmed by dividing it into various variables (Table 4, Figure 2).

There was a statistically significant correlation between the EC exposure level, and the exposure level of OC and TC as confirmed through Pearson’s correlation analysis. Correlation coefficients between EC and OC and between EC and TC were 0.582 (*p* < 0.01) and 0.800 (*p* < 0.01), respectively. Moreover, correlation coefficients varied according to the EC exposure level. When the EC concentration was 10 μg/m^3^ or higher, the correlation coefficients between EC and OC and between EC and TC were 0.874 (*p* < 0.01) and 0.965 (*p* < 0.01), respectively, indicating a statistically significant correlation. However, when the EC concentration was 10 μg/m^3^ or lower, the correlation coefficients between EC and OC and between EC and TC were 0.199 (*p* = 0.171) and 0.446 (*p* <0.01), respectively, showing decreased correlation (Figure 3, Appendix A).

## 4. Discussion

We investigated worktime exposure levels of respirable EC, OC, and TC to which shipyard transporter signal workers may be exposed, and the factors affecting exposure. The GMs (range) of EC, OC, and TC of transporter signal workers were 4.16 (0.69 to 47.81), 15.81 (4.66 to 49.90), and 21.09 (7.28 to 97.71) μg/m^3^, respectively.

We compared the level of exposure to DEE among shipyard transporter signal workers with that of other industries by measuring the exposure level of respirable EC as a surrogate. The exposure level of signal workers was much lower compared with that of underground miners (GM: 62–202 μg/m^3^) and tunnel workers in construction (GM: 87–163 μg/m^3^) [18,19,20,21,22], and was similar to those of train crews and maintenance workers (GM: 3–4 μg/m^3^), surface workers in mining (GM: 2 μg/m^3^), local and long-haul truck, bus, and taxi drivers (GM: 1–9 μg/m^3^), and household-waste collectors (GM: 4.8 μg/m^3^) [15,19,22,23,24,25,26].

The occupational exposure of these ECs was highest in closed spaces. Separation of the source and the workspace reportedly yields low exposure levels [14]. As shipyard transporter signal work is also often performed in the open air, signal workers were split into groups with low EC exposure levels. However, even when working outdoors, the work is performed near the exhaust port of the vehicle. In the case of the transporter, the load capacity is much larger than 5 t, a typical standard for heavy-duty vehicles. According to a study by Seshagiri (2003), in train environments, workers operating at the rear side of an exhaust port face the exhaust gas from the airflow with the vehicle movement, leading to more exposure, compared with workers at the front of the exhaust [27]. In the same way, for transporter signal workers, the workers performing signal work moving with the transporter may have more direct exposure to the exhaust gas moving with the airflow. The fact that the emissions from a diesel vehicle moving outdoors becomes the main source of EC exposure level can also be applied to the case of household-waste collectors, given the similar workspace or the location of the exhaust port in the vehicle and the work position. Job tasks, vehicle movement speed, engine emission standard, the distance between work location and exhaust port, and workload affected the EC exposure level of household-waste collectors [15]. The exposure levels of EC, OC, and TC in the shipyard transporter signal work did not show a significant difference according to variables such as the department of the worker, task, signal work position, number carried, transporter age, and load capacity. Furthermore, unlike other studies that reported higher PM exposure levels in older vehicles, the EC exposure level was higher for transporters less than 10 years of age compared with that of one more than 10 years old. The only variable that showed a significant difference in the EC exposure level was the measurement date, and the variation in exposure level for each measurement date may be affected by meteorological factors. To minimize the effect of meteorological factors, a multiple regression analysis was performed using the results measured in June. Through this, it was confirmed that signal work position, number carried, and load capacity are factors that can affect the EC exposure level for signalmen.

Outdoor exhaust gas concentrations can be affected by environmental and meteorological factors. PM2.5, EC, and OC concentrations increase on days with light winds and a stable atmosphere, and on days with strong winds, the concentration of PM can be reduced through dilution and mixing of air [9,12]. We found no significant differences in the mean exposure to DEE for each variable. The DEE exposure level was lower on the second and third days when the workload was higher, and the vehicle age was older, indicating that exposure to DEE by measurement date was greatly affected by meteorological factors. The wind speed of the first day when the GMs of EC, OC, and TC exposure levels were 13.39, 26.07, and 41.23 μg/m^3^, respectively, was 1.3 m/s. This speed corresponds with light air (class 1), according to the Beaufort wind scale. On the second day when the wind speed was 5.3 m/s and conditions corresponded to a gentle breeze (class 3), the EC, OC, and TC exposure levels were 2.66, 13.28, and 16.41 μg/m^3^, indicating relatively low levels of exposure. On the third day, when the wind speed was 3.3 m/s, corresponding to a light breeze (class 2), the EC, OC, and TC exposure levels were 3.19, 13.87, and 18.03 μg/m^3^, indicating a slightly higher level than that observed on the second day (Figure 1, Appendix A). The prevailing wind for each day of measurement in Figure 1 also shows the differences by day. On days when EC, OC, and TC exposure levels were high, winds from the WNW and WN prevailed, and on the second and third days, wind from the ENE, NE, and ESE prevailed. In the area where the study site was located, the mean daily wind speed in 2019 was equivalent to class 1–3, which is 5.4 m/s or less, on 328 days (approximately 90% of the time). Further investigation of the impact of prevailing winds in a future study would help estimate annual mean DEE exposure levels for shipyard transporter signal workers with large daily weather variations. Follow-up studies considering real-time wind speed and exposure of EC are necessary. We also think that follow-up studies such as real-time wind speed and exposure of EC are necessary.

The concentration of TC is the sum of the concentrations of EC and OC. In studies of DEE exposure of railroad train crews, variables that may have more impact on DEE exposure showed a higher correlation with EC than with OC. Therefore, EC may be a better surrogate of DEE than OC [28]. Moreover, although the sources of EC and OC were the same, EC is generated primarily by combustion [29], while multiple sources other than DEE can generate OC and affect the OC and TC concentrations at the worksite [28]. The ratio of OC/EC in DEE is typically than 1.0, and when the ratio of OC/EC exceeds 2.0, the cause of the additional amount of OC should be considered [29]. The mean OC/EC ratio in this study was 5.27, which was higher than the generally known OC/EC ratio, but the ratio of OC/EC differed depending on DEE exposure level. In samples with an EC concentration of 10 μg/m^3^ or higher, the mean OC/EC ratio was 1.34 (range: 0.88 to 2.36), but for concentrations below 10 μg/m^3^, the mean OC/EC ratio was high, at 6.16 (range: 1.10 to 25.14). The ratio of OC/EC reported for household-waste collection workers was 9.7 (range: 1.4 to 26.1), which was higher than the results in this study. High OC/EC ratios stem from a brief exposure to high concentrations of OC when waste workers open lids of waste receptacles [16]. Most DEE sources in this study came from the exhaust gas of a transporter as the measurements were performed at night when no gasoline engines were running or any other work was going on. The high OC concentration in transporter signal workers may have been caused by exposure to painting or blasting dust with DEE for a short time during the unloading of the ship blocks in places such as painting or blasting booth.

Few countries have set EC as a marker for exposure, though occupational exposure limits (OELs) for DEE have been established. The U.S. Mining Safety and Health Administration set the criteria for TC at 160 μg/m^3^. The American Conference of Governmental Industrial Hygienists proposed a threshold limit value of 20 μg/m^3^ for EC in 2001. There is no national standard in the U.S. for EC, although the state of California recommends a level of 20 μg/m^3^. The OEL is 100 μg/m^3^ for respirable or inhalable EC in Switzerland, Austria, Australia, and New Zealand. In December 2018, the European Union adopted a legal OEL of 50 μg/m^3^ for EC; the regulation will take effect in 2026 in underground mines and tunnel construction and in 2023 in other industries. CAREX Canada recently set a recommended OEL of 20 μg/m^3^ for EC in mining and 5 μg/m^3^ in other industries [15,30,31]. In this study, the mean exposure level in all signal workers was 4.16 μg/m^3^, which corresponds to a low exposure group that includes truck, bus, or taxi drivers and household-waste collectors. However, the levels varied depending on the meteorological factors. No samples exceeded the EU standard of 50 μg/m^3^, but 8.3% of samples (*n* = 5) exceeded the California-recommended standard of 20 μg/m^3^. Moreover, 35.0% of the samples (*n* = 21) exceeded the CAREX Canada recommended standard of 5 μg/m^3^. The EC concentration measured in this study is not high compared to the level at which health effects were shown in mines. Nevertheless, in some conditions, values closed to the OEL in Europe were measured. This is why we should pay attention to the health effects of DEE in this group.

Our study has some limitations. First, DEE exposure of the shipyard transporter signal workers investigated in this study may not be representative of DEE exposure of all the transporter signal workers, considering the small sample size and large variability in outdoor meteorological conditions. There appears to be an association between the work and meteorological conditions, especially wind. The probability of the level exceeding half or more of the exposure standard depending on weather condition is high when compared with the OEL for EC used in the EU, although an OEL does not exist in South Korea. Second, we did not consider the effect of personal protective equipment. Measures are needed to enable the use of personal protective equipment. Use of such equipment is currently not practical as the job involves whistles and radios. Third, we did not clearly show the quantitative change of diesel exhaust exposure considering special weather such as rainy periods. However, fine particles, such as diesel particles, have lesser washout effects of atmospheric particulate by rainfall than coarse particles [32].

## 5. Conclusions

This is the first study to measure DEE exposure characteristics of signal workers of a ship-block transporter in a shipyard using EC, OC, and TC as surrogate. The objectives of this study were to identify the DEE exposure level of signal workers and establish the factors that can affect exposure in shipyard transporter signal workers. We confirmed that the most influential factor was meteorological conditions. After minimizing the effects of the climate, it was confirmed that the signal work position, number carried, and the load capacity were factors affecting the EC exposure of the signalman. The mean exposure level of EC was low; signal work involves exposure to high concentrations of EC depending on the wind speed and prevailing wind. However, due to the nature of the work, there is considerable daily variation in exposure levels, and it is difficult to determine the EC exposure level of individuals using measurements of general, occupational environments. Further daily monitoring of meteorological conditions and EC concentrations that allow for estimates of annual cumulative EC exposure will foster protection of workers’ health. We also recommend establishing an OEL and reduction controls against exhaust sources to prevent diseases caused by DEE.

## Figures and Tables

**Figure 1 ijerph-17-04398-f001:**
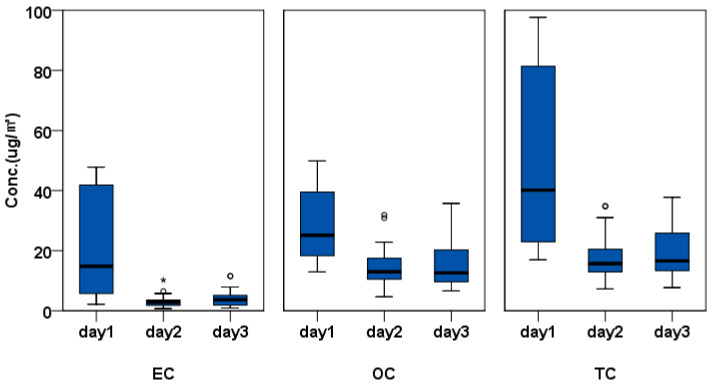
Exposure level of elemental carbon (EC), organic carbon (OC), and total carbon (TC) by date of measurement. The box represents the interquartile (IQ) range which contains the middle 50% of the records and whiskers indicate 10th and 90th percentiles. Outliers (circles, ◦) are cases with values between 1.5 and 3 times the IQ range. Extremes (asterisks, *) are cases with values more than 3 times the IQ range.

**Figure 2 ijerph-17-04398-f002:**
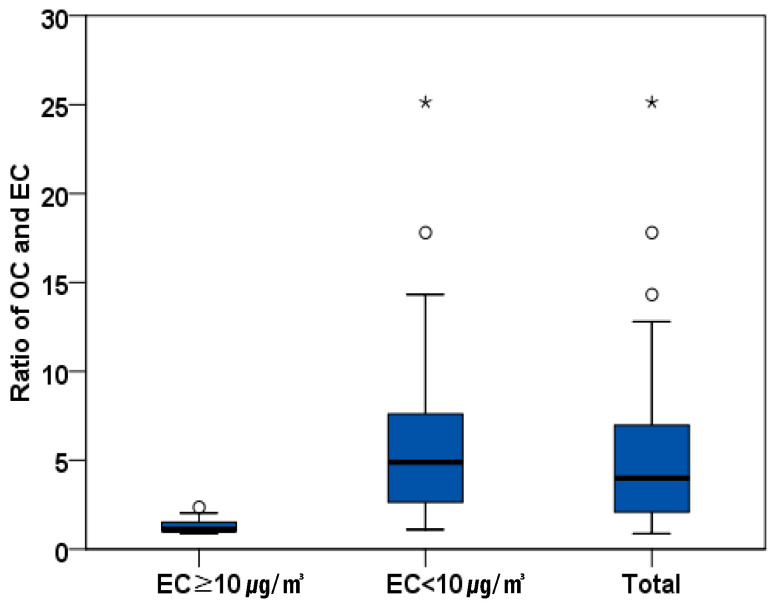
Ratio of organic carbon (OC) and elemental carbon (EC) by EC exposure level. The box represents the interquartile (IQ) range which contains the middle 50% of the records and whiskers indicate 10th and 90th percentiles. Outliers (circles, ◦) are cases with values between 1.5 and 3 times the IQ range. Extremes (asterisks, *) are cases with values more than 3 times the IQ range.

**Figure 3 ijerph-17-04398-f003:**
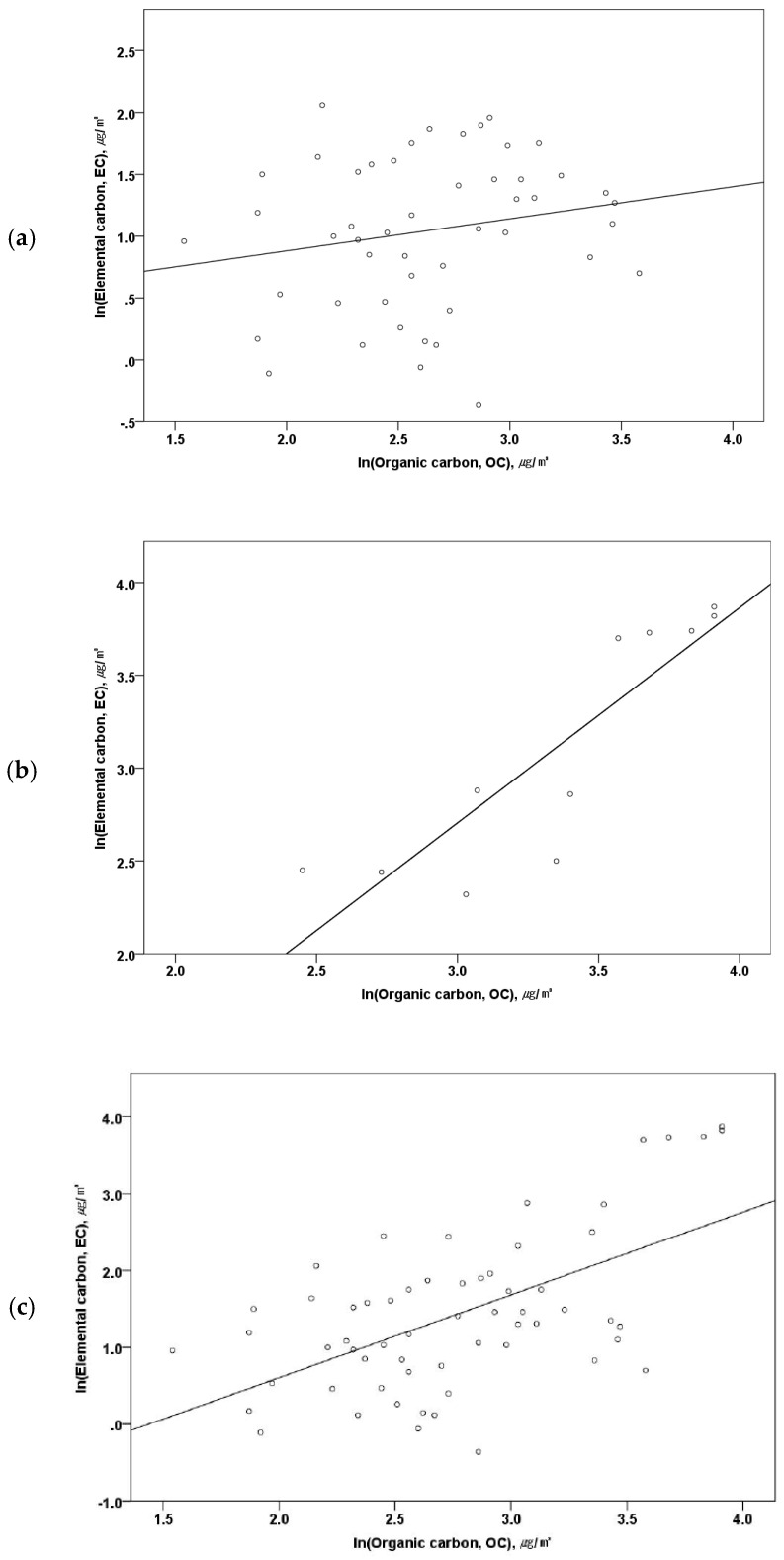
Correlation of EC and OC by EC exposure level: (**a**) EC < 10 μg/m^3^ (r = 0.199, *p* = 0.171), (**b**) EC ≥ 10 μg/m^3^ (r = 0.874, *p* < 0.01), (**c**) EC data all (r = 0.582, *p* < 0.01).

**Table 1 ijerph-17-04398-t001:** Characteristics of transporters and the number carried by measurement day.

No.	Loading Capacity, t	Production Year	Transporter Age, Year	Body Weight, t	Body Length, m	Body Width, m	Number Carried		
							May	June	
							Day1	Day2	Day3
1	300	2003	16	75	21	5	-	28	40
2	350	2002	17	73.5	21	6	-	18	21
3	350	2002	17	73.5	21	6	17	22	34
4	350	2016	3	75	21	6	20	-	-
5	500	2007	12	80	21	5	-	26	26
6	500	2016	3	80	21	5	27	-	-
7	600	2008	11	92	21	7.5	19	-	-
8	600	2012	7	121	21	7.5	-	25	22
9	1000	2006	13	160	21	10	4	5	8

**Table 2 ijerph-17-04398-t002:** Exposure levels of elemental carbon (EC), organic carbon (OC), and total carbon (TC) in transporter signal work according to variables.

Variables	N	EC, μg/m^3^			OC, μg/m^3^			TC, μg/m^3^		
		GM (GSD)	Range	*p*-Value	GM (GSD)	Range	*p*-Value	GM (GSD)	Range	*p*-Value
Total	60	4.16 (2.76)	0.69–47.81	-	15.81 (1.73)	4.66–49.90	-	21.09 (1.85)	7.28–97.71	-
Month										
May	14	13.39 (2.90)	2.15–47.81	<0.001	26.07 (1.59)	12.95–49.90	<0.001	41.23 (1.91)	16.95–97.71	<0.001
June	46	2.91 (1.99)	0.69–11.56		13.57 (1.62)	4.66–35.75		17.20 (1.54)	7.28–37.75	
Department										
Part1	44	4.36 (3.05)	0.89–47.81	0.559	15.07 (1.74)	6.51–49.90	0.268	20.52 (1.95)	7.70–97.71	0.564
Part2	16	3.65 (1.99)	0.69–11.50		18.02 (1.70)	4.66–32.22		22.77 (1.55)	7.28–35.78	
Task										
Signalman	48	4.49 (2.52)	0.89–45.73	0.371	15.08 (1.71)	4.66–49.82	0.386	20.52 (1.79)	7.28–95.55	0.784
Forklift	7	2.50 (3.85)	0.69–42.04		20.17 (1.67)	12.32–45.93		24.03 (1.96)	13.61–87.98	
Uniloader	5	4.03 (4.04)	1.59–47.81		17.62 (2.08)	9.27–49.90		22.89 (2.47)	10.86–97.71	
Signal work position										
Front	31	3.84 (2.51)	0.94–45.73	0.115	15.19 (1.76)	4.66–49.82	0.898	19.97 (1.83)	7.28–95.55	0.667
Rear	17	5.97 (2.42)	0.89–41.83		14.88 (1.62)	6.52–39.53		21.57 (1.75)	7.70–81.36	
Number carried										
<25	27	4.14 (3.21)	0.90–45.73	0.454	14.60 (1.78)	4.66–49.82	0.639	19.87 (2.01)	7.28–95.55	0.649
≥25	21	4.99 (1.59)	2.34–17.47		15.72 (1.62)	6.51–32.22		21.39 (1.49)	9.81–47.35	
Transporter age, year										
<10	12	7.97 (2.66)	2.34–41.83	0.011	16.96 (1.67)	9.10–39.53	0.385	25.70 (1.95)	11.82–81.36	0.124
≥10	36	3.71 (2.30)	0.89–45.73		14.50 (1.72)	4.66–49.82		19.04 (1.72)	7.28–95.55	
Load capacity, t										
<500	22	4.40 (3.05)	0.89–41.83	0.945	17.36 (1.80)	4.66–39.53	0.247	23.25 (1.93)	7.28–81.36	0.402
<1000	18	4.72 (1.62)	2.15–17.47		13.35 (1.52)	6.51–29.88		18.36 (1.46)	9.81–47.35	
≥1000	8	4.23 (3.29)	1.50–45.73		13.47 (1.77)	7.19–49.82		18.70 (2.05)	8.90–95.55	

**Table 3 ijerph-17-04398-t003:** Multiple regression analysis results to predict natural log-transformed EC levels.

Independent Factors		Unstandardized Coefficients		Standardized Coefficients	*t*	*p*-Value	F	*p*-Value
		B	Standard Error	Beta				
(Constant)		−1.571	0.599		−2.623	0.013	9.070	<0.001
Department	Part1	Reference	0.219	0.308	1.994	0.055		
	Part2	0.437						
Signal work position	Front	Reference	0.173	0.398	3.262	0.003		
	Rear	0.564						
Number carried		0.058	0.015	0.811	3.854	0.001		
Load capacity, t		0.002	0.001	0.766	3.735	0.001		

**Table 4 ijerph-17-04398-t004:** Ratio of OC/EC by EC exposure level and variables.

Variables	EC ≥ 10 μg/m^3^				EC < 10 μg/m^3^				Total			
	N	Mean	SD	Range	N	Mean	SD	Range	N	Mean	SD	Range
Total	11	1.34	0.49	0.88–2.36	49	6.16	4.63	1.10–25.14	60	5.27	4.59	0.88–25.14
Measurement date												
Day 1	8	1.29	0.50	0.88–2.36	6	3.93	1.80	2.25–6.90	14	2.42	1.79	0.88–6.90
Day 2	1	2.03	-	-	22	6.47	5.10	1.78–25.14	23	6.28	5.07	1.78–25.14
Day 3	2	1.34	0.23	1.00–1.33	21	6.47	4.66	1.10–17.80	23	6.01	4.70	1.00–17.80
Department												
Part1	9	1.26	0.48	0.88–2.36	35	5.62	3.95	1.48–17.80	44	4.73	3.95	1.48–17.80
Part2	2	1.68	0.50	1.33–2.03	14	7.49	5.98	1.10–25.14	16	6.77	5.91	1.10–25.14
Task												
Signalman	9	1.40	0.52	0.88–2.36	39	5.04	3.18	1.10–14.32	48	4.36	3.21	0.88–14.32
Forklift	1	1.09	-	-	6	12.91	7.46	5.65–25.14	7	11.22	8.14	1.09–25.14
Uniloader	1	1.04	-	-	4	6.90	3.82	3.85–12.48	5	5.73	4.22	1.04–12.48
Signalman position												
Front	4	1.22	0.35	0.88–1.71	27	5.72	3.45	1.48–14.32	31	5.14	3.56	0.88–14.32
Rear	5	1.53	0.63	0.95–2.36	12	3.53	1.78	1.10–7.61	17	2.94	1.78	0.95–7.61
Number of Transport												
<25	7	1.36	0.59	0.88–2.36	20	6.05	3.65	1.78–14.32	27	4.84	3.77	0.88–14.32
≥25	2	1.52	0.27	1.33–1.71	19	3.98	2.22	1.10–9.03	21	3.75	2.24	1.10–9.03
Transporter Age, Year												
≤10	4	1.19	0.38	0.88–1.71	8	3.05	1.11	2.16–4.88	12	2.43	1.29	0.88–4.88
>10	5	1.56	0.60	1.00–2.36	31	5.56	3.35	1.10–14.32	36	5.00	3.41	1.00–14.32
Load Capacity, t												
<500	6	1.46	0.60	0.88–2.36	16	6.96	3.56	1.10–14.32	22	5.46	3.93	0.88–14.32
<1000	1	1.71	-	-	17	3.18	1.43	1.48–6.90	18	3.10	1.43	1.48–6.90
≥1000	2	1.05	0.06	1.00–1.09	6	5.23	2.95	2.25–10.23	8	4.18	3.16	1.00–10.23

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
