# Peer review of "Characteristics of Occupational Exposure to Diesel Engine Exhaust for Shipyard Transporter Signal Workers"

_ijerph, 2020, doi:10.3390/ijerph17124398_

Round 1

Reviewer 1 Report

Overall comments: Thank you for the opportunity to review the manuscript entitled “Characteristics of exposure to diesel engine exhaust for shipyard transported signal workers”. This study investigates the exposure to shipyard workers who perform signal work for a heavy-duty transporter to diesel emission exhaust. The authors showed that average exposure was low and daily exposure was dependant of meteorological factors such as wind speed. As these workers are exposed to diesel emission exhaust from a ship transporter which are not subject to heavy-vehicle regulations due to the operational workload, it is important to determine the diesel emission exhaust load and levels of exposure. Therefore, this study is important and of interest to the readers of IJQRPH and I am recommending this manuscript for publication with a minor revision. One limitation of the study was the collection of data over three days and were limited to one season. This seems quite limited. I ask the authors to provide justification for this. Also, a minor comment, please review the sentence on page 12 line 303. Is DEE typically less that 1.0 or more then 1.0?

Author Response

Dear reviewers and editor:

We are grateful for the insightful comments from you. We have edited our manuscript according to your comments, to add more clarity to our findings and to draw more interests of your journal’s potential readers. We have highlighted the corrected parts by yellow band. We would like to express my sincere gratitude to you. Thank you.

Reviewer #1

  1. Overall comments: Thank you for the opportunity to review the manuscript entitled “Characteristics of exposure to diesel engine exhaust for shipyard transported signal workers”. This study investigates the exposure to shipyard workers who perform signal work for a heavy-duty transporter to diesel emission exhaust. The authors showed that average exposure was low and daily exposure was dependent of meteorological factors such as wind speed. As these workers are exposed to diesel emission exhaust from a ship transporter which are not subject to heavy-vehicle regulations due to the operational workload, it is important to determine the diesel emission exhaust load and levels of exposure. Therefore, this study is important and of interest to the readers of IJERPH and I am recommending this manuscript for publication with a minor revision. One limitation of the study was the collection of data over three days and were limited to one season. This seems quite limited. I ask the authors to provide justification for this.

Answer) Thank you for your comments. According to your suggestion, we added a limitation and justification for the limitation in discussion part. Considering the measurement in the general industrial hygiene field, authors thought that the measurement time of 3 days is reasonable. Through this study, we were able to confirm that transporter signal work, which is performed outdoors, is affected by meteorological factor (in particular, wind speed). Since the three-day wind speed included in the measurement includes about 90% of the annual average wind speed in 2019 in the study area, we think it is appropriate to confirm the EC exposure characteristics of the transporter signal work from the data in this study. However, follow-up studies considering real-time wind speed and exposure of EC are necessary. This was highlighted line 315-321 on page 11.

  1. Also, a minor comment, please review the sentence on page 12 line 303. Is DEE typically less than1.0 or more than 1.0?

Answer) The OC/EC ratio is typically close to either “1” or “<1” in occupational environments where diesel vehicles are the only source of EC (Lee et. al., 2016). The OC/EC ratio was 1.1 for automobile exhaust (Watson et al., 2001) and 9.0 for biomass combustion (Cachier et al., 1989; Cao et al., 2005). Higher OC/EC ratio is less affected by motor vehicle exhaust and it means there may be other secondary factors. The OC/EC ratio is generally known as less than 1 for diesel engines and 1 or higher for gasoline engines (Turpin and Huntzicker, 1991; Lin and Tai, 2001). In this study, we interpreted the high OC concentration like the following. “The high OC concentration in transporter signal workers may have been caused by exposure to painting or blasting dust with DEE for a short time during the unloading of the ship blocks in places such as painting or blasting booth.”

Reviewer 2 Report

The present paper reports on an Industrial Hygiene study dealing with the exposure assessment and characterization of TC, EC and OC in workers performing signal work for a heavy-duty shipyard transporter occupationally exposed to diesel engine exhaust (DEE). The main goal id to understand factors affecting such an exposure pathway and pattern, investigating, besides all the operational issues along a significant period, also meteo-climatic changes.

The study design is appropriate. The analytical methods are well standardized. The results are widely described and analyzed to account for the internal and external variable accounting for the observed changes in the exposure pattern. However, considering that more than on factor at a time seems to affect exposure, a multivariare analysis could better describe the weigth of influence of each independent variables on exposure. Maybe that some metereological variables show a co-collinearity, thus being interrelated among them.

The discussion seems consistent with the results. The main conlusion is tat the occupational exposure of these ECs was highest in closed spaces and that there is a clear incluence of meteorological conditions on DEE exposure of transporter signal workers who work outdoors. These findings are rather observative and apparently obvious; however, they can be useful to estimate the risk of expsure and of possible health effects in this cathegory of workers, as compared to other working conditions.

The paper suffers from other minor issues that however should be faced to improve the manuscript and also attract the interest of readers.

Table 3 (Exposure level of elemental carbon (EC), organic carbon (OC), and total carbon (TC) by meteorological factors and task is very difficult to read in its present format. I suggest to draw some graphics considering only the statistically significant variables.

Similarly, I suggest to consider the possibility to convert some huge tables in pictures.

Although the different kind of operations are well described, the paper lack to investigate the relationships between changes in exposure and other climatic variables (eg., it is unclear whether workink activities have been performed during raining periods, whith the obvious consequence of a falls in particulate concentration)

Author Response

Dear reviewers and editor:

We are grateful for the insightful comments from you. We have edited our manuscript according to your comments, to add more clarity to our findings and to draw more interests of your journal’s potential readers. We have highlighted the corrected parts by yellow band. We would like to express my sincere gratitude to you. Thank you.

Reviewer #2

1. The study design is appropriate. The analytical methods are well standardized. The results are widely described and analyzed to account for the internal and external variable accounting for the observed changes in the exposure pattern. However, considering that more than one factor at a time seems to affect exposure, a multivariate analysis could better describe the weight of influence of each independent variable on exposure. Maybe that some meteorological variables show a co-collinearity, thus being interrelated among them.

Answer) Thanks for your comments. As you suggested, multiple regression analysis was performed and it has been added in the manuscript (abstract, method, result, discussion, and conclusion) according to the analysis results. The exposure level of EC is considered to be significantly affected by weather conditions such as wind speed. In order to identify variables that influence the exposure level of EC, multiple regression analysis was performed using the results measured in June, when the effect of wind was relatively small. The selected variables are signal work position, number carried, and load capacity. The results are shown in Table 3 on page 7.

2. The discussion seems consistent with the results. The main conclusion is that the occupational exposure of these ECs was highest in closed spaces and that there is a clear influence of meteorological conditions on DEE exposure of transporter signal workers who work outdoors. These findings are rather observative and apparently obvious; however, they can be useful to estimate the risk of exposure and of possible health effects in this category of workers, as compared to other working conditions.

Answer) Thank you for your comments. According to your suggestion, we added the comments like the following in discussion part. This was highlighted line 355-358 on page 11.

The EC concentration measured in this study is not high compared to the level at which health effects were shown in mines. Nevertheless, in some conditions, values closed to the OEL in Europe were measured. This is why we should pay attention to the health effects of DEE in this group.

3. Table 3 (Exposure level of elemental carbon (EC), organic carbon (OC), and total carbon (TC) by meteorological factors and task is very difficult to read in its present format. I suggest to draw some graphics considering only the statistically significant variables.

Answer) As suggested, authors inserted Figure 1 on page 6 instead of Table 3 to make it easier for readers to see.

4. Similarly, I suggest to consider the possibility to convert some huge tables in pictures.

Answer) In addition, Figure 2 is presented in addition to Table 4 on page 8 so that the results can be seen at a glance.

5. Although the different kind of operations are well described, the paper lack to investigate the relationships between changes in exposure and other climatic variables (eg., it is unclear whether working activities have been performed during raining periods, with the obvious consequence of a falls in particulate concentration)

Answer) The work according to the climatic situation is only stopped on a particularly strong day, such as a typhoon. Signal work is performed even on rainy days. Fine particles, such as diesel particles, have less the washout effects of atmospheric particulate by rainfall than coarse particles (Guo L.-C. et al. 2016). However, the limitation of this study was that it did not clearly show the quantitative change of diesel exhaust exposure. We added the limitation in discussion part. This was highlighted line 367-370 on page 12.

Round 2

Reviewer 2 Report

The Authors have properly addressed the few issues raised. By my side, the manuscript deserves publication.